

# A long-term record of blended satellite and in situ sea surface temperature for climate monitoring, modeling and environmental studies

V. Banzon[1], T. M. Smith[2], C. Liu[1,3] and W. Hankins[1,3]

[1]NOAA National Centers for Environmental Information (NCEI), 151 Patton Ave., Asheville, NC 28801, USA
[2]NOAA/STAR/SCSB/ESSIC University of Maryland, College Park, MD 20740, USA
[3]Earth Resources Technology, 14401 Sweitzer Lane Ste. 300, Laurel, MD 20707

*Correspondence to*: V. Banzon (viva.banzon@noaa.gov)

**Abstract.** This paper describes a blended sea-surface temperature (SST) dataset that is part of the National Oceanic and
Atmospheric Administration (NOAA) Climate Data Record (CDR) Program product suite. Using optimum interpolation (OI), in situ and satellite observations are combined on a daily and 0.25° spatial grid to form an SST analysis, i.e., a spatially complete field. A large-scale bias adjustment of the input infrared SSTs is made using buoy and ship observations as a reference. This is particularly important for the time periods when volcanic aerosols from the El Chichon and Mt. Pinatubo eruptions are widespread globally. The main source of SSTs is the Advanced Very High Resolution Radiometer (AVHRR),
available from late 1981 to the present, which is also the temporal span of this CDR. The input and processing choices made to ensure a consistent dataset that meets the CDR requirements is summarized. A brief history and an explanation of the forward production schedule for the preliminary and science-quality final product is also provided. The dataset is produced and archived at the newly formed National Centers for Environmental Information (NCEI) in Network Common Data Form (netCDF) at http://doi.org/doi:10.7289/V5SQ8XB5.

**1 Introduction**

Sea surface temperature (SST) is an essential climate variable (ECV). The Global Climate Observing System (GCOS) project developed a list of ECV's to focus worldwide observation efforts on a limited set of variables that are climate-relevant, technically feasible, and cost-effective (Bojinski et al., 2014). Collectively, ECVs can help develop adaptation and mitigation strategies, assess risks, allow attribution and prediction, and support climate services. SST is useful for monitoring
El Niño events and multi-decadal ocean changes. It is also relevant to quantification and modeling of many other aspects of climate such as air-sea interaction, ocean acidification to determine solubility of carbon dioxide, biophysical processes, marine organism distributions, etc. However, models require not just observations but complete data fields, also referred to as analyses. Today, satellites offer high spatial and temporal coverage and are therefore, the main source of SST



observations. Additional processing is applied to satellite data to form analyses to allow for bias corrections and gap-filling, and thereby increase spatio-temporal consistency.

The objective of this work is to describe the NOAA ¼ ° daily Optimum Interpolation SST version 2 (or dOISST.v2, herein), an analysis that has been selected by the National Oceanic and Atmospheric Administration (NOAA) Climate Data Record

(CDR) program as an operational CDR. This implies that the dOISST.v2 meets the definition of CDR put forward by the National Research Council (2004): it is of sufficient length, consistency and continuity to determine climate variability. Furthermore, operational NOAA CDRs undergo a research to operations process to ensure systematic production and quality assessment, thereby increasing the dataset maturity in aspects like transparency, usability and data preservation following metadata standards. A preliminary assessment using the maturity matrix of Bates and Privette (2012) indicated that

dOISST.v2 has a high maturity in both science and applications, but needed improvements in accessibility and transparency to users. As part of the effort to address this deficiency, this paper describes the dOISST.v2 CDR dataset, in the context of its historical beginnings and evolution, its current temporal and spatial characteristics, dataset formats and access, as well as provides examples of applications. Much of this information is publicly available but has not been summarized in a single document.

## 2 Historical background


Here, precursors to the dOISST.v2 that have evolved into the current CDR are briefly reviewed to highlight the original motivation and subsequent modifications. Historically, the widely-used name "Reynolds SST" has been applied to all current and precursor products, and is therefore ambiguous and not used here. Reynolds (1988) first introduced the concept of a blended SST analysis that takes advantage of the sea-truth offered by in situ data and the high coverage of satellite data.

Prior to 1980, ships were the only source of observations, and the spatial-temporal coverage was sufficient only for a coarse-scale analysis. Starting in late 1981, satellite-based SST observations became available daily from an infrared instrument, the Advanced Very High Resolution Radiometer (AVHRR), with Global Area Coverage (GAC) resolution at ~4 km. Using high quality drifting buoys as reference, Reynolds (1988) found that monthly analyses, based on AVHRR SSTs alone or blended with in situ data, were slightly more accurate than one based on in situ data alone (with drifter data withheld).

However, notable satellite biases occurred under specific situations, e.g., at cloud edges or in the presence of volcanic aerosols. Large differences between satellites and buoys (>1 °C) were reported following the Mt Pinatubo and El Chichon eruptions (Reynolds, 1993; Reynolds, 1988). For these periods, reliable global SST fields could be produced if in situ data was used to benchmark the satellite data to form blended analyses (Reynolds, 1988; Reynolds and Marisco, 1993). This large-scale benchmarking is also referred to as a "satellite bias adjustment".

Reynolds and Smith (1994) adopted the optimum interpolation (OI) method to increase the effective resolution of the blended analysis to 1°, and the temporal frequency from monthly to weekly, and later to daily. This was the first time the name OISST was used. Along the marginal ice zone where observations were sparse, the interpolation was relaxed to the



freezing point of seawater (-1.8 °C). This was slightly modified in the follow-up version 2 (also referred to as OI.V2 in Reynolds et al., 2002), where a regression equation was used to estimate proxy SSTs from sea ice concentrations. NCEP continues to produce the weekly OI.V2 for seasonal forecasting, but users should be aware that there are important differences relative to the current CDR, i.e., the dOISST.v2, as discussed in Banzon et al. (2014).

## 3 The climate data record

Reynolds et al. (2007) increased the daily OISST resolution to the current ¼° grid using a bias correction scheme that employs Empirical Orthogonal Teleconnections (EOT) modes, rather than the Poisson method used in OI.V2. The use of EOTS also had the additional advantage that it allowed estimation of the bias error. Moreover, the earlier OISST analyses used operational (i.e., near real time) satellite data, but for the ¼° daily OISST, higher quality input datasets reprocessed from the start of the mission were used preferentially over operational data. Most significant was the AVHRR Pathfinder reprocessing, which improved SST retrievals using nighttime buoy data to compute a revised set of coefficients for each NOAA satellite (e.g., Kilpatrick et al., 2001). However, the cold bias associated with the eruptions of El Chichon and Mt. Pinatubo remained a challenge even in the more recent AVHRR SST reprocessing efforts. Another minor change was that the proxy SST calculation was restricted closer to the ice margin, i.e., where sea-ice concentrations exceed 50%, to avoid potentially erroneous SST estimates in the more open waters, where the fit was much noisier.

Reynolds et al. (2007) referred to the above product as AVHRR-only, in reference to the source of satellite SSTs. The same paper describes a companion analysis that uses the same methodology but includes data from the Advanced Microwave Scanning Radiometer on the Earth Observing System (AMSR-E). This product, called AVHRR+AMSR, is not part of this CDR, due to the short record (2002 to 2011) of AMSR-E data. It should be noted that the Climate Forecast System (CFS) at NCEP uses the same Reynolds et al. (2007) methodology to generate their intial SST fields (Saha et al., 2010; 2014), but may use different inputs (e.g., both infrared and microwave satellite SSTs) over time, and therefore will differ from this CDR.

The daily OISST was upgraded to version 2 primarily to increase temporal and spatial smoothing (Reynolds, 2009). This involved changing parameter settings but the core methodology described in Reynolds et al. (2007) remained the same. The treatment of in situ data was also slightly modified. Historically, in situ observations were predominantly from ships. In the 21[st] century, more accurate buoy data had become increasingly dominant over ship data, providing a better reference temperature. A constant (~0.14 °C) was subtracted from the ship data to compensate for the global-average ship-buoy difference (Reynolds et al., 2010; Reynolds, 2009). Modern ship measurements tend to be warmer because they typically use intake samples that can be warmed when taken into the ship engine room. However, there is much scatter in individual differences and better understanding of ship bias is needed to reduce the uncertainty in this correction. The net effect of this adjustment is that the daily OISST tends to be cooler than the weekly OI.V2 particularly in the 1980s and 1990s when ship data was dominant, making the long-term trend slightly steeper.





### 3.1 Dataset description

The daily OISST is available in netCDF and binary (FORTRAN IEEE big-Endian) formats. In this paper, the archived netCDF files, publicly available at the National Centers for Environmental Information (NCEI) website, are described. However, the same data are repackaged and distributed elsewhere for specific projects or organizations such as the Group for

High Resolution SST (or GHRSST) and Observations for Model Intercomparison Projects (Obs4MIPs), with accompanying metadata and documentation, but are not described here. The heritage binary format will be eventually phased out.

A single netCDF file contains four global gridded fields (1440 X 720) pertaining to one day. The primary variable is the analyzed SST (units in °C; Fig.1 a).  Since buoys are used as a reference, this is sometimes referred to as a "bulk" SST, at a nominal buoy depth of 1 m. The SST input data types (AVHRR daytime, AVHRR nighttime, buoys, ships, proxy SST; Table

1) are first averaged to 1/4° superobservations. The in situ data (consisting of the buoy and adjusted ship data) are collectively used to make large-scale adjustments to satellite data using the EOT modes.  All data are merged during the interpolation, using the pre-computed error characteristics as weights. More details can be found in Reynolds et al. (2007). Grid points corresponding to land, permanent ice shelves, and most inland waters are not processed and assigned a missing value of -999.


Three other gridded fields at the same 1/4° spatial resolution complement the daily analysis:

- Anomalies (i.e., the daily OISST minus the 1971-2000 climatological mean; units in ° C; Fig. 1 b)) represent departures from "normal" or average conditions. The anomalies are provided so users can easily compute climate indices, such as the NINO3.4 (Fig. 2a). The 1971-2000 climatology is partly based on an in situ analysis for the

years that satellite data are not available (1971-1981), and on the weekly OISST for years satellite data are available from 1982 onward (Xue et al., 2003).  A climatological mean computed from daily OISST (1982-2011) is now available and is more suitable to use with this dataset, as explained in Banzon et al. (2014). It will be used in the next version.  User should consider that with a long-term warming, a more recent period may produce a warmer climatological mean, and thus when subtracted from the analyzed SSTs, it produces cooler anomalies.

- The standard error (with units in °C; Fig. 1 c) provides a measure of uncertainty in the estimated SST, allowing users to exclude (using a threshold) or to minimize (using weights) the importance of grid point values with greater errors, as needed for the specific application, e.g., resource management, risk analysis, or assimilation into a model.

- The seven-day median of daily sea ice concentrations (expressed as a real fraction from 0.0 to 1.0; Fig. 1 d) is the basis of the proxy SST estimate in the marginal ice zone.  Aside from reducing noise, the temporal median

populates the time series in the early 1980s when satellite sea ice observations were available only every other day. There is a data gap between Dec 4, 1997 to Jan 14, 1998, for which there are no sea ice data.  This field is effectively also an ice mask if the user opts to exclude areas with high ice concentrations.



The input datasets to dOISST.v2 are listed in Table 1 and, and have been evaluated in more detail in Reynolds et al. (2007). While reprocessed inputs are used whenever possible, only operational datasets meet the low latency needs of the daily updates. Users should be aware that sensor problems are typically cannot be addressed in near-real time. The release date of the dOISST.v2 was Nov 2008.. To minimize the impact of near real time sensor problems, data from two AVHRRs are used from 2007 onward (Table 2).

The analysis for the first day in the record used climatology as a first guess. For all other days the previous analysis is used as a first guess. For the daily update, a 1-day delayed analysis is produced. Two weeks later after more data has become available, the analysis is repeated to produce higher quality "final" product. The final and preliminary runs can be identified in the global attributes of the netCDF file, and the preliminary filename also contains the word" preliminary". Only the "final" product is archived.

### 3.2 Basic characterization

The daily OISST is available for the full period of record from September1981 to the present. The dataset is similar to other global daily SST analyses in that monthly, seasonal, multi-year averages can be computed on global, regional, and local scales. For climate applications, the daily OISST is unique because it extends from late 1981 to the present, and therefore spans over 30 years, often cited as the minimum period needed to distinguish inter-annual variations from long-term variations. The characteristic seasonal SST cycle, represented here by the1982-2011 climatological mean, varies by location. In the tropics, it is exemplified by the NINO3.4 region (Figure 1a), where the seasonal signal is weak. The start of the 1997 El Nino event is marked by SSTs that are more than one standard deviation greater than the climatological mean for over three months. A stronger seasonal cycle occurs in the temperate zone, as seen in the Gulf of Maine (Fig. 2b). The SSTs over the entire year 2012 exceed the climatological mean plus one standard deviation. The daily progression shows particularly elevated May-June temperatures, which initiated a season of anomalous lobster catch (Mills et al., 2013). Of course, these atypical events can also investigated by examining the anomalies.

Long time series are ideal for computing multi-decadal trends. On an annual scale, the 1982 to 2014 global linear trend using dOISST is ~0.12 °C per decade. The wintertime trend is slightly smaller (0.09 °C per decade using only January monthly averages; Fig 3a) than in summertime (0.14°C per decade using only July monthly averages; Fig 3b). The ¼° resolution data allows trends to be computed on more local scales but comparisons should always be made with in situ measurements, if available. The daily SST information can be used to generate other climate-relevant parameters. For example, the number of days that SSTs are above a threshold, also known as degree-days, is an indicator of thermal stress for corals. In fact, many ecological responses to a changing ocean can be modeled in terms of the cumulative effect of daily temperatures on growth, reproduction, recruitment, and the like.

Global validation of the dOISST using buoy and ship data is not an independent assessment because in situ data is used to make the product, although the amount of satellite data incorporated is much greater. Comparisons with other SST analyses





would have the same issue since most analyses also use in situ data. With that caveat, Reynolds and Chelton (2010) showed that relative to buoys, the dOISST.v2 and other analyses all exhibit regional variability in performance, reflecting their methodological differences. For the dOISST.v2, the root-mean-square error relative to the buoys is about 0.3 °C. Reynolds and Chelton (2010) also found that the product degrades in quality during prolonged periods of no data, e.g., seasonally

cloud-covered areas such as the Gulf Stream in winter. Analyses that included microwave data had better results when infrared retrievals were not available, because of the added data coverage. But when infrared data were available, the addition of microwave data reduced the quality of the resulting analyses because microwave SSTs are less accurate. In any case, an analysis is not necessarily the best source of SST for a single point in space or time because it is a smoothed product. The advantage is that where there are no observations, an analysis provides interpolated values, and over time, a

consistent long-term record. It should be noted that the analysis is also useful as a reference field for identifying bad data and is therefore used in several satellite algorithms for quality control. It also serves as a first guess or as ancillary data for computing parameters that require a known temperature field.

A potential validation reference is Argo data, which are not used as an input to dOISST. In fact, most SST analyses do not use Argo, as agreed on by the GHRSST community to have a common independent validation dataset. However, Argo

observations are available only since 2003, are limited in number and are located deeper (~4 m) than other buoy measurements. An assessment of dOISST.v2 will be the subject of future work. Certainly, as the Argo dataset grows, it might be possible to withhold only a portion for validation, and use the rest in the analysis. For a future version of dOISST, Argo data could improve data coverage in areas with sparse ship and buoy data such as the Tropical Pacific, where moored buoy data was degraded for some years.

In terms of feature resolution, i.e., the ability of an analysis to reproduce mesoscale ocean features, Reynolds et al. (2007) showed dOISST performs well. While some analyses are available at higher resolution spatial grids, Reynolds and Chelton (2010) showed they do not necessarily provide good feature resolution and small-scale artifacts may also be present. Obviously, the higher resolution products are also limited by the availability of data at that resolution.

## 4 Conclusions

A long-term sea-surface temperature climate data record consisting of in situ and satellite data blended daily on a ¼° grid is available for climate monitoring, modeling, validation, and a wide range of other applications. The dataset uses AVHRR data from 1981 to the present, bias-adjusted relative to situ data. This produces a time series that is more consistent than satellite infrared retrievals alone. This CDR is produced, distributed and generated by the National Centers for Environmental Information (NCEI), a newly formed entity that is a merger of three NOAA data centers including the

National Climatic Data Center (NCDC), which originally produced this dataset.

Compared to the precursor weekly OISST at NCEP, the CDR has many updates including higher spatial resolution, reprocessed inputs, and adjustment of ship data to match buoys. The CDR is also used as an ancillary field in reprocessed



and operational satellite algorithms including the Pathfinder AVHRR SST, Tropical Rainfall Measuring Mission (TRMM) rain rate, and Aquarius salinity. The CDR version of the dOISST.v2 is available in netCDF format.

**Author Contribution**

All authors contributed to the text. V. Banzon wrote the draft, with significant text added by T. Smith. Processing details
were provided by C. Liu and B. Hankins, who run the operational production.

**Acknowledgements**

The authors would like to thank R. W. Reynolds (retired), H.-M. Zhang, and G. Peng for providing comments that greatly improved this paper.  Members of the OISST Integrated Products team, including C. Hutchins, P. Jones, R. McFadden, V. Toner and D. Wunder, helped meet CDR program requirements for transition and maintenance of dOISST.v2.

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



Table 1. Input datasets to the daily OISST. The data sources are explained in detail in Reynolds et al. (2007).

| Input Type | Reprocessed or Higher Quality Datasets | Operational Datasets |
|---|---|---|
| Satellite (AVHRR SSTs) | Pathfinder 5.0/5.1(1981-2006) | Navy (2007-present) |
| In situ SSTs | ICOADS 5.1 (1981-2006) | NCEP (2007-present) |
| Sea ice concentrations | GSFC NASA (1981-2004) | NCEP (2005-present) |



Table 2. Platform time spans of AVHRR inputs to the daily OISST. Note that two satellites at a time are used beginning
January 2007.

| Dataset | Start date | End date | Platform | Sensor |
|---|---|---|---|---|
| Pathfinder | 1981 Aug 24 | 1985 Jan 03 | NOAA-7 | AVHRR/2 |
| | 1985 Jan 04 | 1988 Nov 07 | NOAA-9 | AVHRR/2 |
| | 1988 Nov 08 | 1994 Sep 13 | NOAA-11 | AVHRR/2 |
| | 1994 Sep 14 | 1995 Jan 21 | NOAA-9 | AVHRR/2 |
| | 1995 Jan 22 | 2000 Oct 11 | NOAA-14 | AVHRR/2 |
| | 2000 Oct 11 | 2002Dec 31 | NOAA-16 | AVHRR/3 |
| | 2003 Jan 01 | 2005 Jun 04 | NOAA-17 | AVHRR/3 |
| | 2005 Jun 05 | 2006 Dec 31 | NOAA-18 | AVHRR/3 |
| Navy | 2006 Jan 01 | 2008 Dec 31 | NOAA-17 | AVHRR/3 |
| | 2007 Jan 01 | 2011 Aug 15 | NOAA-18 | AVHRR/3 |
| | 2009 Jan 01 | Present | MetOP-A | AVHRR/3 |
| | 2011 Aug 16 | Present | NOAA-19 | AVHRR/3 |



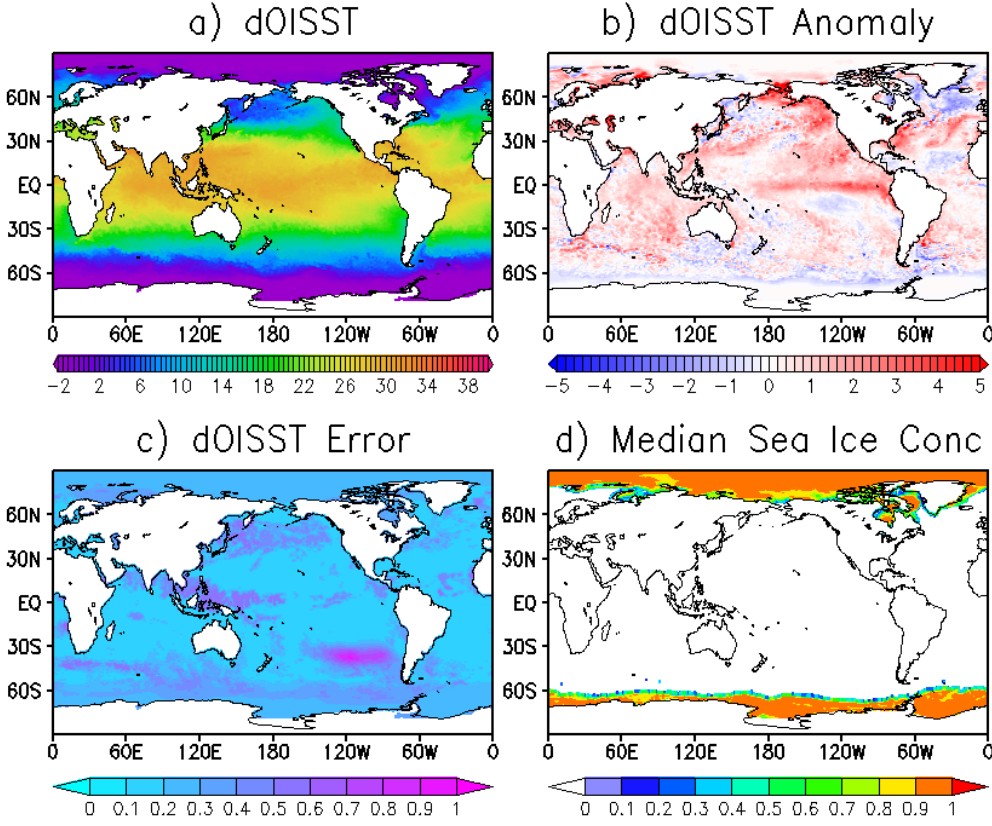

Figure 1. Examples of the four variables in a singles file: a) dOISST.v2, b) dOISST anomaly, c) Errror and d) Median sea ice concentrations. Data are shown for 20 Jun 2015.



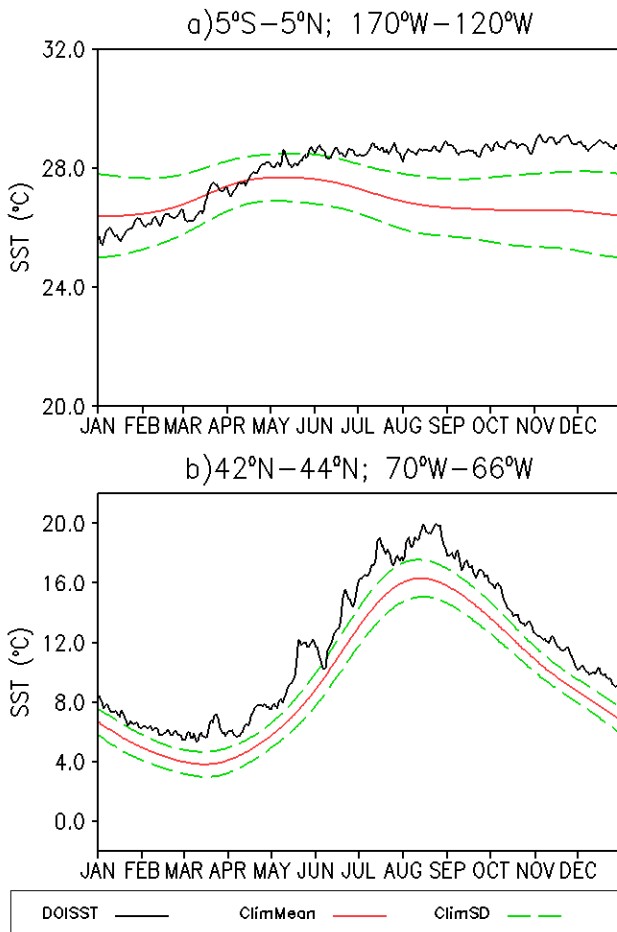

Figure 2. a) Temporal progression of the 1997 daily OISST in the NINO3.4 region (black line), and the 1982-2011 climatological mean (red line) for the same area. The green lines are offset from the mean by plus and minus one standard deviation, and shows characteristic variability. b) Same but in 2012 in the Gulf of Maine (after Mills et al., 2013). Titles show coordinates used to compute the area weighted means.



a)

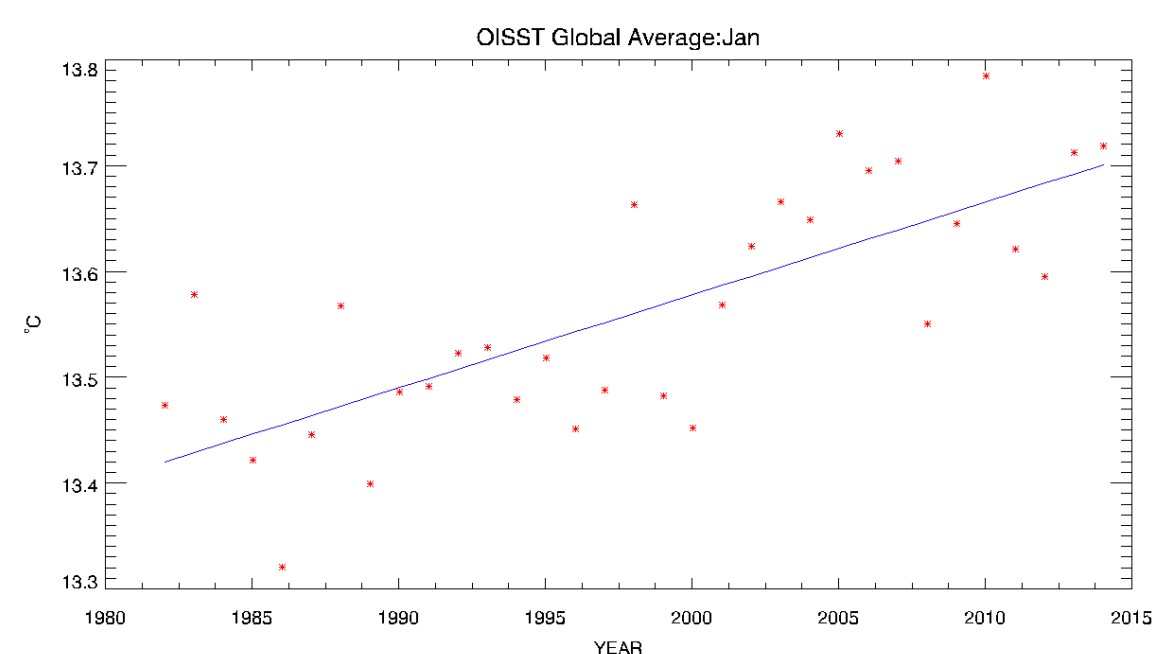

b)

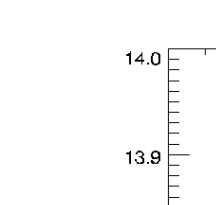

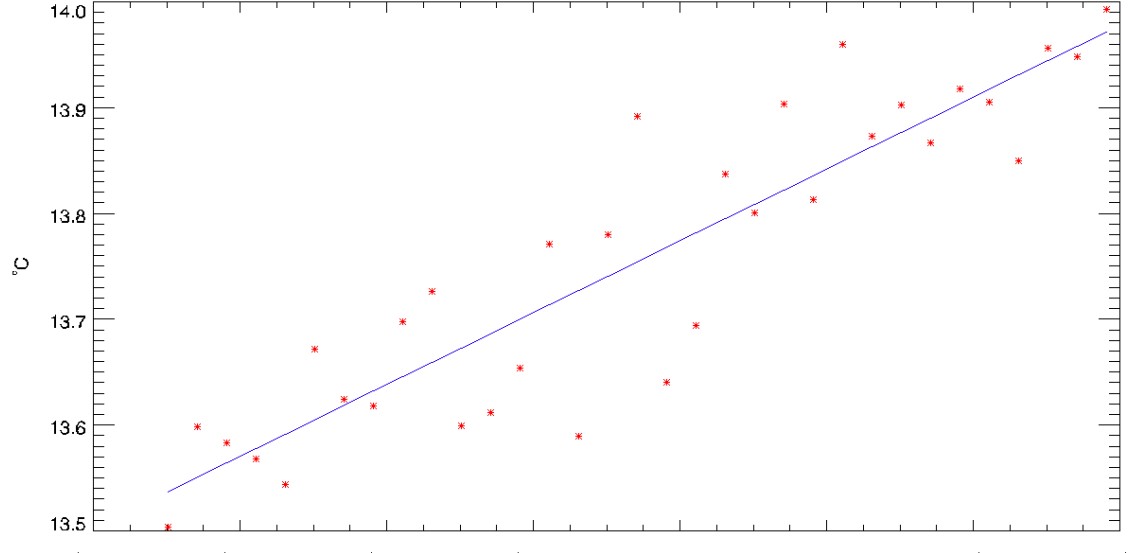

Figure 3. Global OISST trends (1982-2014) using a) January monthly means only, b) July monthly means only