# Peer review of "A long-term record of blended satellite and in situ sea surface temperature for climate monitoring, modeling and environmental studies"

_Earth System Science Data, 2015_

## Short Comment (SC1) · 6 Feb 2016

The paper describes a new higher resolution of the historical Reynolds Sea Surface Temperature Data Set. Additionally, it is a much needed summary of the historical processing of the Reynolds Optimally Interpolated Sea Surface Temperature Data Sets. The paper is well written and organized with a introduction of the Reynolds OI and the methodology used. The summary of how the in-situ data is used in the analysis and the impact is extremely valuable.

Strengths: Very well written paper. Having the summary of the Reynolds OI documented in one paper is extremely valuable. The discussion of the impacts of the advancement of the in-situ data used and the impact on the analysis is extremely helpful.
[Figure]

Weaknesses: The authors make some strong statements which should be clarified. For example biases are not only caused by the volcanic eruptions but also by other sources of aerosol, such as the Saharan Dust Storms. These references should be added to the document. Towards the end the authors also mention: "While some analyses are available at higher resolution spatial grids, Reynolds and Chelton (2010) showed they y do not necessarily provide good feature resolution and small scale artifacts may also be present.

. Obviously, the higher resolution products are also limited by the availability of data at that resolution." This statement needs to be clarified before publishing. The higher resolution does provide good feature resolution within the constraints of the availability of infrared data. For example the Multi-Scale Ultra-High Resolution Sea Surface Temperature Data (MUR) set incorporates a methodology which prserves the high resolution when infrared data is available, but a smoother product when only the lower resolution microwave data is available. The advantage of higher resolution has been clearly seen. Some references to add:

J. Vazquez-Cuervo and E. Armstrong, The Effect of Aerosols and Clouds on the Retrieval of Infrared Sea Surface Temperatures, Journal of Climate, 2004, (11), 3921-3933.

Vazquez-Cuervo, J., B. Dewitte, T. M. Chin, E. M. Armstrong, S. Purca, E. Alburqueque, 2013: An Analysis of SST Gradients off the Peruvian Coast: The impact of going to higher resolution. Remote Sensing of the Environment, vol (131). 76-84.

---

## Author Comment (AC1) · 11 Feb 2016

Thank you to Dr. Vazquez for the excellent comment. Yes, it is true that dust aerosols and clouds are also a source of biases and we will definitely add that with the proper citation in the revised version. The impact of volcanic aerosols was emphasized because it has such a widespread effect. The statement about high resolution will be clarified in the revision and the suggested paper will be cited. In addition, it came to our attention that Dr. Vazquez has a publication on Pathfinder version 5.0 and find that relevant to this paper. We will add that to the citations.

---

## Referee Comment (RC2) · B. Buongiorno Nardelli (Referee) · 29 Feb 2016

This paper provides a nice summary of the historical evolution of the SST L4 fields produced by NOAA. It then focuses on the most recent dOISST.v2 described in Banzon et al. (2014), providing examples of application for interannual/climatological studies. In fact, this product covers more than 30 years, going back to late 1981 and it has been identified by NOAA as an operational Climate Data Record. The paper is very clearly written and organized. However, I feel that the discussion on products' effective space-time resolution could be strengthened (e.g. by adding a comparative illustration of wavenumber spectra obtained from the different global analyses availabe, even at higher nominal resolution, as those developed in the framework of the European

Space Agency Climate Change Initiative). As these aspects are quite relevant, clear messages would help scientific users to identify which product better fits their needs.

---

## Referee Comment (RC3) · B. Buongiorno Nardelli (Referee) · 3 Mar 2016

As I said in my previous comment, scientific users would surely benefit having some information to identify which product better fits their needs. So, I think that even providing a single additional figure, as you propose, would be informative and interesting. I suggested the ESA CCI for comparison because it represents somehow the European counterpart of NOAA CRD. On the other hand, I understand that working on an evolving product is not an optimal choice and thus agree that taking the OSTIA REP product distributed through the European CMEMS service would be fine. Other products probably do not cover such large periods but possibly include many different sources of data, so comparison even with ultra-high-resolution L4 could probably help highlighting

the advantages and limitations of each dataset. As you remark,this is however beyond the scope of the present paper, so I would leave it to your choice whether to include also this part or not.

---

## Author Comment (AC3) · 3 Mar 2016

Thank you for the enlightened comments. The spectral analysis is an interesting point, but beyond the scope of the paper, which is intended only to describe the dataset and does not do any new analysis. But the comment is well taken, and we agree that there is a need for a spectral comparison of many SST datasets.

I am aware of work in progress by Mike Chin at JPL, whom I contacted. A few years ago, he agreed to do spectral analyses of existing SST analyses submitted to GHRSST. Unfortunately he has not completed the work, but said he might be able to send me a figure with the spectra of our product and a high resolution one for comparison, or even OSTIA, produced by the MetOffice. So while I would rather not add that to the paper,

[Figure]

I could add the figure offered by Mike Chin. Would this be a satisfactory compromise? I'd have to add him as an author and add some text. Please let me know what is the preferred resolution.

Just for completeness, I also contacted Chris Merchant who leads the CCI SST team. He sent me a paper which describes the current CCI product, which was developed in phase 1, but is still being further developed in phase 2. It extends from 1991 to 2010, and there are plans to extend backward to 1982. A future paper could make the comparison against this future dataset. Because this dataset is still in development, Mike Chin has not had a chance to look at this CCI dataset.

---

## Referee Comment (RC4) · B. Buongiorno Nardelli (Referee) · 14 Mar 2016

Proposed modifications are fine for me.

———————————————

---

## Author Comment (AC4) · 14 Mar 2016

The paper is not an analysis paper so no methodological description will be provided. Here is the proposed text to replace page 6, paragraph beginning with line 20:

In terms of feature resolution, i.e., the ability of an analysis to reproduce mesoscale ocean features, Reynolds et al. (2007) showed that dOISST performs well. While some SST analysis products use higher resolution grids than dOISST, Reynolds and Chelton (2010) have demonstrated that a higher grid resolution does not necessarily mean that more higher-resolution SST features are captured in the analysis. To illustrate this point, the power spectral densities of three SST analysis datasets are shown in Figure 4. These datasets are three of those examined by Reynolds and Chelton (2010),

but the latest versions of the datasets (from the first 2 months of 2016) are used in the plot presented here. The spectra are smoother because they are the average of several areas rather than a single area, in order to provide a global representation of each dataset. The three products differ in inputs and methodologies but the main focus here is that they differ in grid resolutions: dOISST is on a 1/4 degree grid; the Operational SST and Sea Ice Analysis (OSTIA) is on a 1/20 deg grid (Donlon et al., 2012), and theRemote Sensing Systems (RSS) analysis is computed on a 1/11 deg grid (http://podaac.jpl.nasa.gov/dataset/MW IR OI-REMSS-L4-GLOB-v4.0). The figure shows that the SST feature resolution is very similar between dOISST and OSTIA even if the latter uses a finer grid size.
* * *
[Figure]

Fig. 1. Power spectra of various analyzed SST fields from the first two months of 2016. All spectra indicate some large scale (and perhaps seasonal) SST patterns (wavelengths larger than 1500 km) identically.